Transcription factor organic cation transporter 1 (OCT-1) affects the expression of porcine Klotho (KL) gene

Li Yan
Wang Lei
Zhou Jiawei
Li Fenge lifener@mail.hzau.edu.cn
Key Laboratory of Pig Genetics and Breeding of Ministry of Agriculture & Key Laboratory of Agricultural Animal Genetics, Breeding and Reproduction of Ministry of Education, Huazhong Agricultural University , Wuhan , PR China
Verger Alexis
Electronic publication date: 2016 Jul 14
Publication date: 2016
Volume: 4
Electronic Location ID: e2186
Received 2016 Mar 1; Accepted 2016 Jun 7
Copyright: ©2016 Li et al.
Copyright year: 2016
Copyright holder: Li et al.
License: This is an open access article distributed under the terms of the Creative Commons Attribution License, which permits unrestricted use, distribution, reproduction and adaptation in any medium and for any purpose provided that it is properly attributed. For attribution, the original author(s), title, publication source (PeerJ) and either DOI or URL of the article must be cited.
License URL: https://creativecommons.org/licenses/by/4.0/

Keywords: KL gene, OCT-1, Pig, MARC0022311

Funding: Key Projects in National Science R&T Program 2015BAD03B02 2014BAD20B01 Hubei Science R&T Program 2014BBB008 2014BBA194 Fundamental Research Funds for the Central Universities This work was supported financially by Key Projects in National Science R&T Program (2015BAD03B02, 2014BAD20B01), Hubei Science R&T Program (2014BBB008, 2014BBA194), and Fundamental Research Funds for the Central Universities. The funders had no role in study design, data collection and analysis, decision to publish, or preparation of the manuscript.

==============================
Klotho (KL), originally discovered as an aging suppressor, is a membrane protein that shares sequence similarity with the β-glucosidase enzymes. Recent reports showed Klotho might play a role in adipocyte maturation and systemic glucose metabolism. However, little is known about the transcription factors involved in regulating the expression of porcine KL gene. Deletion fragment analysis identified KL-D2 (−418 bp to −3 bp) as the porcine KL core promoter. MARC0022311SNP (A or G) in KL intron 1 was detected in Landrace × DIV pigs using the Porcine SNP60 BeadChip. The pGL-D2-A and pGL-D2-G were constructed with KL-D2 and the intron fragment of different alleles and relative luciferase activity of pGL3-D2-G was significantly higher than that of pGL3-D2-A in the PK cells and ST cells. This was possibly the result of a change in KL binding ability with transcription factor organic cation transporter 1 (OCT-1), which was confirmed using electrophoretic mobility shift assays (EMSA) and chromatin immune-precipitation (ChIP). Moreover, OCT-1 regulated endogenous KL expression by RNA interference experiments. Our study indicates SNP MARC0022311 affects porcine KL expression by regulating its promoter activity via OCT-1.

Introduction

The Klotho (KL) gene encodes a membrane protein that shares a sequence similarity with the β-glucosidase genes and its product may function as part of a signaling pathway that regulates aging and morbidity in age-related diseases (Ko et al., 2013). Mutant mice lacking the KL gene shows multiple aging disorders and a shortened life span (Kuro-o et al., 1997). KL−∕− mice have the pattern of ectopic calcification certainly contributed by the elevated phosphate and calcium levels (Hu et al., 2011; Ohnishi et al., 2009). KL also acts as a deregulated factor of mineral metabolism in autosomal dominant polycystic kidney disease (Mekahli & Bacchetta, 2013). Mice that lacked Klotho activity are lean owing to the reduced white adipose tissue accumulation, and are resistant to obesity induced by a high-fat diet (Ohnishi et al., 2011; Razzaque, 2012).

KL expression is regulated by thyroid hormone, oxidative stress, long-term hypertension and so on (Koh et al., 2001). Some transcription factors such as peroxisome proliferator-activated receptor gamma (PPAR-γ) also can regulate KL expression (Zhang et al., 2008). A double- positive feedback loop between PPAR-γ and Klotho regulates adipocyte maturation (Chihara et al., 2006; Zhang et al., 2008). Briefly, chromatin immuno-precipitation (ChIP) and gel shift assays find a PPAR-responsive element within the 5′-flanking region of human KL gene. Additionally, PPAR-γ agonists increases KL expression in HEK293 cells and several renal epithelial cell lines, while the induction is blocked by PPAR-γ antagonists or small interfering RNAs (Zhang et al., 2008). Furthermore, Klotho can induce PPAR-γ synthesis during adipocyte maturation (Chihara et al., 2006). However, little is known about the transcription factors involved in regulating the expression of porcine KL gene.

Several hundreds of thousands of porcine SNPs were discovered using next generation sequencing technologies, and Illumina Inc used these SNPs, as well as others from different public sources, to design a high-density SNP genotyping assay (Ramos et al., 2009). SNP MARC0022311 is one 64,232 SNPs on the Porcine SNP60K BeadChip. In our study, we detected MARC0022311 SNP in some pigs using 60K SNP chip and found that this SNP could affect the transcriptional regulation of KLOTHO gene. To investigate the transcriptional regulation of porcine KL gene, we identified the core promoter of porcine KL gene, analyzed its upstream regulatory elements and revealed that transcription factor OCT1 directly bound to the core promoter region of porcine KL gene and regulated its expression.

Materials and Methods

Ethics statements

All animal procedures were performed according to the protocols approved by the Biological Studies Animal Care and Use Committee of Hubei Province, PR China. Sample collection was approved by the ethics committee of Huazhong Agricultural University (No. 30700571 for this study).

MARC0022311 polymorphism in pigs

Nineteen Landrace × DIV crossbred pigs were genotyped with the Porcine SNP60 BeadChip (Illumina) using the Infinium HD Assay Ultra protocol, which was conducted under the technical assistance by Compass Biotechnology Corporation. DIV was a synthetic dam line derived by crossing Landrace, Large White, Tongcheng or Meishan pigs. Raw data had high genotyping quality (call rate >0.99) and were analyzed with the GenomeStudio software.

In silico sequence analysis

KL gene sequence ENSSSCG00000009347 was available on the ENSEMBL online website (http://asia.ensembl.org/index.html). We obtained the up-stream sequence of porcine KL gene for promoter prediction. The potential promoter was analyzed using the online neural network promoter prediction (NNPP) (http://www.fruitfly.org/seq_tools/promoter.html) and Promoter 2.0 prediction server (http://www.cbs.dtu.dk/services/Promoter/). Transcription factor binding sites were predicted using biological databases (BIOBASE) (http://www.gene-regulation.com/pub/programs.html) with a threshold of 0.90 and TFsearch with a threshold of 85 (Akiyama, 1995). Threshold represents minimum probability of predicted transcription factor.

Cell culture, transient transfection and luciferase assay

The porcine kidney (PK) cells and swine testis (ST) cells obtained from China Center for Type Culture Collection (CCTCC) were cultured at 37 °C in a humidified atmosphere of 5% CO2 using DMEM supplemented with 10% FBS (Gibco).

Four KL promoter deletion fragments (KL-D1: −178 bp to −3 bp, KL-D2: −418 bp to −3 bp, KL-D3: −599 bp to −3 bp and KL-D4: −835 bp to −3 bp) were cloned into pGL3-Basic vector to determine the core promoter region. The plasmids contained pig KL intron 1 fragments (g.1474 A and g.1474 G) followed by KL-D2 promoter (−418 bp to −3 bp) were reconstructed, then transfected using lipofectamine 2000 (Invitrogen) into PK cells and ST cells. Plasmid DNA of each well used in the transfection containing 0.8 µg of KL promoter constructs and 0.04 µg of the internal control vector pRL-TK Renilla/luciferase plasmid. The enzymatic activity of luciferase was then measured with PerkinElmer 2030 Multilabel Reader (PerkinElmer).

RNA interference

Double-stranded small interfering RNAs (siRNAs) targeting OCT-1 were obtained from GenePharma. Cells were co-transfected with 2 µl of siRNA, 0.2 µg of reconstructed plasmids using Lipofetamine 2000™ reagent for 24 h. Transfection mixtures were removed, and fresh complete DMEM medium was added to each well. Finally, the enzymatic activity of luciferase was then measured with PerkinElmer 2030 Multilabel Reader (PerkinElmer).

Quantitative real time PCR (qPCR)

qPCR was performed on the LightCycler® 480 (Roche) using SYBR® Green Real-time PCR Master Mix (Toyobo). Primers used in the qPCR were shown in Table 1. qPCR conditions consisted of 1 cycle at 94 °C for 3 min, followed by 40 cycles at 94 °C for 40 s, 61 °C for 40 s, and 72 °C for 20 s, with fluorescence acquisition at 74 °C. All PCRs were performed in triplicate and gene expression levels were quantified relatively to the expression of β-actin. Analysis of expression level was performed using the 2−ΔΔCt method (Livak & Schmittgen, 2001). Student’s t-test was used for statistical comparisons.

Table 1 Primers and DNA oligos used in this study.

Primer	Primer sequence (5′-3′)	Amplicon length (bp)	Tm (°C)	
5′-Bio of A (+)	GGTAATGTTGTAATAATGGCTAA		60	
5′-Bio of A (−)	TTAGCCATTATTACAACATTACC		
cold probe of A (+)	GGTAATGTTGTAATAATGGCTAA		60	
cold probe of A (−)	TTAGCCATTATTACAACATTACC		
cold probe of G (+)	GGTAATGTTGTAATAGTGGCTAA		60	
cold probe of G (−)	TTAGCCACTATTACAACATTACC		
KL_ChIP_ PF	TGAAGACCACTGCTACACACTT		59	
KL_ChIP_ PR	AGCAAACAGGTTTTGTGGAGC		
KL_ D1_ PF	CGGGGTACCTTGTTGGATGTTTTGTTTGTCTAGCTAGC	193	58	
KL_ D_ PR	CGACGCGTCCCTGTGAAGGCTTGTTT	
KL_ D2_ PF	CGGGGTACCTATGAGGAGGTGGGTTGGCTAGCTAGC	433	59	
KL_ D_ PR	CGACGCGTCCCTGTGAAGGCTTGTTT	
KL_ D3_ PF	CGGGGTACCCACTTAACCTCTTATTCTTGAGTTACTAGCTAGC	614	59	
KL_ D_ PR	CGACGCGTCCCTGTGAAGGCTTGTTT	
KL_ D4_ PF	CGGGGTACCACATAAAAGTTAGAAAATCAGAGAACTAGCTAGC	850	59	
KL_ D_ PR	CGACGCGTCCCTGTGAAGGCTTGTTT	
OCT1_ qPCR_ PF	TGAACAATCCGTCAGAAACC	196	58	
OCT1_ qPCR_ PR	TGAGCAGCAGCCTGTAAACT	
KL_ qPCR_ PF	ACCCGTATTTATTGATGGAGAC	173	57	
KL_ qPCR_ PR	GGAACTTCATCTGAGGGTCTAA	
KL_ intron1_ ChIP_ PF	GCCGTAGATAATTGAAGC	130	50	
KL_ intron1_ ChIP_ PR	TCTGTGGTAGCAAACAGG	
KL_ intron2_ ChIP_ PF	GCCAGTGTAAGGTGTTACC	114	51	
KL_ intron2_ ChIP_ PR	ATTCTCCAAAGAAGACATACA	
KL_ intron3_ ChIP_ PF	CAAGATTGTACCGTGGAG	171	50	
KL_ intron3_ ChIP_ PR	GGTCATTTGACATCATTCT	
Notes.

Protective bases and induced enzyme sites were in italic and bold, respectively.

Western blotting

Western blotting was performed as described previously (Tao et al., 2014). Five µg proteins were boiled in 5 × SDS buffer for 5 min, separated by SDS-PAGE, and transferred to PVDF membranes (Millipore). Then, the membranes were blocked with skim milk and probed with anti-KL (ABclonal). β-actin (Santa Cruz) was used as a loading control. The results were visualized with horseradish peroxidase-conjugated secondary antibodies (Santa Cruz) and enhanced chemiluminescence.

Electrophoretic mobility shift assays (EMSA)

Nuclear protein of PK and ST cells was extracted with Nucleoprotein Extraction Kit (Beyotime). The oligonucleotides (Sangon) corresponding to the OCT-1 binding sites of KL intron 1 (Table 1) were synthesized and annealed into double strands. The DNA binding activity of OCT-1 protein was detected by LightShift® Chemiluminescent EMSA Kit (Pierce). Ten µg nuclear extract was added to 20 fmol biotin-labeled double stranded oligonucleotides, 0.1 mM EDTA, 2.5% Glycerol, 1× binding buffer, 5 mM MgCl2, 50 ng Poly (dI⋅dC) and 0.05% NP-40. In addition, control group added 2 pmol unlabeled competitor oligonucleotides, while the super-shift group added 10 µg OCT-1 antibodies (Santa Cruz). The mixtures were then incubated at 24 °C for 20 min. The reactions were analyzed by electrophoresis in 5.5% polyacrylamide gels at 180 V for 1 h, and then transferred to a nylon membrane. The dried nylon was scanned with GE ImageQuant LAS4000 mini (GE-Healthcare).

Chromatin immunoprecipitation (ChIP) assay

ChIP assays were performed using a commercially available ChIP Assay Kit (Beyotime) as previously described (Tao et al., 2015). Briefly, after crosslinking the chromatin with 1% formaldehyde at 37 °C for 10 min and neutralizing with glycine for 5 min at room temperature, PK and ST cells were washed with cold PBS, scraped and collected on ice. Then, cells were harvested, lysed and treated by sonication. Nuclear lysates were processed 20 times for 10 s with 20 min intervals on ice water using a Scientz-IID (Scientz). An equal amount of chromatin was immune-precipitated at 4 °C overnight with at least 1.5 µg of OCT-1 antibody (Santa Cruz) and normal mouse IgG antibody (Millipore). Immune-precipitated products were collected after incubation with Protein A + G coated magnetic beads. The beads were washed, and the bound chromatin was eluted in ChIP elution buffer. Then the proteins were digested with Proteinase K for 4 h at 45 °C. The DNA was purified using the AxyPrep PCR Cleanup Kit (Axygen). The DNA fragment of OCT-1 binding sites in KL intron 1 was amplified with the specific primers (Table 1). The PCR procedure was executed with 36 rounds and in the linear range, ChIP assay had 3 biological replicates.

Statistical analysis

Statistical analyses based on two-tailed Student’s t-tests were performed using the Statistical Package for the Social Sciences software. Significance was determined at a 95% confidence interval. All data were expressed as the mean ± standard deviation (S.D.).

Results

MARC0022311 status in pigs

MARC0022311 in KL intron 1 appeared a polymorphism (A or G) in 19 Landrace × DIV pigs, with 12 AA pigs and AG pigs genotyped using the Illumina PorcineSNP60 chip (Data S1) . The SNP (MARC0022311) in pig KL intron 1 was renamed as KL g.1474 A > G according to the standard mutation nomenclature (Den Dunnen & Antonarakis, 2000).

Figure 1 Deletion analysis of pig KL promoter.

(A) Schematic diagram of KL promoter, MARC0022311 (KL g.1474 A > G) and OCT-1 binding site in intron 1. (B) Promoter activities of a series of deleted constructs determined by luciferase assay. The luciferase reporter construct containing the individual sequence KL-D1–KL-D4 (KL-D1:−178–−3 nt, KL-D2:−418–−3 nt, KL-D3:−599–−3 nt and KL-D4: −835–−3 nt) was transfected into ST cells and PK cells, and dual luciferase assays were performed 24 h after transfection. Firefly luciferase activity was normalized to the corresponding Renilla luciferase activity. Values are expressed as means ± SE of three replicates. ***P < 0.001.

Identification of promoter region of the porcine KL gene

An 833 bp contig in 5′ flanking region of pig KL gene was obtained by PCR. To determine the promoter region, four promoter deletions (KL-D1, KL-D2, KL-D3 and KL-D4) were cloned into fluorescent vector based on the prediction of NNPP online software and Promoter 2.0 (Fig. 1A). Luciferase activity analysis in both PK and ST cells revealed that KL-D2 (−418 bp to −3 bp) was essential for its transcriptional activity and was defined as the KL promoter region (Fig. 1B).

MARC0022311 SNP affects the KL expression

Intron SNPs could not change the amino acid sequence, but might alter gene promoter activity by affecting the binding ability of transcription factors (Van Laere et al., 2003). The plasmids contained the wild-type A (g.1,474 A) or mutant G (g.1474 G) sequence followed by KL-D2 were named as pGL3-D2-A and pGL3-D2-G, respectively. Then recombinant DNA fragments were inserted in the downstream of the luc+ gene between the KpnI and HindIII sites. Results showed that luciferase activity of pGL3-D2-G was significantly higher than pGL3-D2-A in both PK cells (P < 0.05) and ST cells (P < 0.01) (Fig. 2A), and indicated that MARC0022311 SNP changed the binding ability of certain regulatory elements affected KL promoter activity.

Figure 2 MARC0022311 in pig KL intron 1 affected promoter activity in PK and ST cells.

(A) Luciferase assays of reporter constructs using pig KL-D2 promoter and intron 1 fragments (g.1474 A and g.1474 G). (B) Luciferase detection after co-transfection of OCT-1 siRNA with pGL3-D2-A and pGL3-D2-G in PK cells. (C) Luciferase detection after co-transfection of OCT-1 siRNA with pGL3-D2-A and pGL3-D2-G in ST cells. *P < 0.05. **P < 0.01. The pGL3-basic was used as the negative control.

The SNP (MARC0022311) located in the first intron of KL gene (+1, 474 bp) was predicted to change the binding ability of OCT-1 by BIOBASE and TFsearch (Fig. S1). After silencing OCT-1 using siRNAs in PK and ST cells, luciferase activity of pGL3-D2-G was significantly lower than pGL3-D2-A (P < 0.05) (Figs. 2B and 2C). Furthermore, compared with the negative control, the luciferase activity of pGL3-D2-A was significantly decreased (P < 0.05) (Figs. 2B and 2C). Thus, MARC0022311 regulated the promoter activity via OCT-1.

However, inhibition of OCT-1 expression significantly suppressed KL expression in PK and ST cells (P < 0.05) (Fig. 3), possibly because OCT-1 could stimulate KL expression by binding KL gene at other sites.

Figure 3 OCT-1 up-regulated KL expression by RNAi.

(A) PK cells were treated with 2 µl OCT-1 siRNA and 2 µl NC for 24 h. Knockdown of OCT-1 was confirmed by qPCR. KL mRNA and protein expressions were analyzed by qPCR and Western blotting. (B) ST cells were treated with 2 µl OCT-1 siRNA and 2 µl NC for 24 h. Knockdown of OCT-1 was confirmed by qPCR analysis. KL mRNA and protein expressions were analyzed by qPCR and Western blotting. *P < 0.05. **P < 0.01. Relative mRNA expression was relative to the expression of β-actin.

Transcription factor OCT-1 binds to the KL intron 1 both in vitro and in vivo

To address whether KL intron 1 contained OCT-1 binding sites in vitro, we used two oligonucleotides (A allele and G allele oligonucleotides) as porcine OCT-1 probes in EMSA. EMSA revealed a highly specific interaction with allele A oligonucleotide, and a 100 fold excess of mutant allele G oligonucleotide could not outcompete the interaction (Fig. 4A). A super-shift was obtained when nuclear extracts from PK and ST cells were incubated with OCT-1 antibodies, providing further biochemical evidence for the presence of OCT-1 in vitro (Fig. 4A). We found the KL genotype at g.1474 A > G locus was AA in PK and ST cells by PCR-sequencing, indicating the endogenous binding of OCT-1 to KL in above two cell lines (Fig. S2). The chromatin was immune-precipitated using an OCT-1 antibody and DNA fragments of the expected size were used as a template to perform PCR amplification. ChIP analysis showed that OCT-1 interacted with KL intron 1 (Fig. 4B). These results showed that transcription factor OCT-1 bound to KL intron 1 both in vitro and in vivo.

Figure 4 Binding of OCT-1 with KL intron 1 was analyzed by EMSA and ChIP.

(A) The probe was incubated with nuclear extract in the absence or presence of 100-fold excess of various competitor probes (mutant or non-labeled probe) or anti-OCT-1. The specific super-shift (DNA-protein-antibody complex) bands were both observed in PK and ST cells. The sequences of various probes were demonstrated under the panel. (B) ChIP assay of OCT-1 binding to the KL intron 1 in PK cells and ST cells. The interaction of OCT-1 in vivo with KL intron region was determined by chromatin immunoprecipitation analysis. DNA isolated from immune-precipitated material was amplified by PCR to amplify KL fragement. Total chromatin was used as the input. Normal mouse IgG was used as a negative control.

Discussion

KL gene encodes a type-I membrane protein that is related to beta-glucosidases (Ko et al., 2013). KL may function as part of a signaling pathway that regulates morbidity in age-related diseases such as atherosclerosis and cardiovascular disease, and mineral metabolism diseases such as ectopic calcification (Ko et al., 2013; Kuro-o et al., 1997; Hu et al., 2011; Ohnishi et al., 2009). Overexpression of KL in the preadipocyte 3T3-L1 cell line can induce expression of several adipogenic markers, including PPARγ, CCAAT/enhancer binding protein alpha (C/EBPα) and CCAAT/enhancer binding protein delta (C/EBPδ), and facilitate the differentiation of preadipocytes into mature adipocytes (Chihara et al., 2006). Eliminating KL function from mice results in the generation of lean mice with almost no detectable fat tissue, and induces a resistance to high-fat-diet-stimulated obesity (Razzaque, 2012; Ohnishi et al., 2011).

Here we found the SNP MARC0022311 located in KL intron 1 in the tested pigs (Data S1). A number of SNPs are proved to have major effects on the phenotypic variations (Markljung et al., 2009; Milan et al., 2000; Ren et al., 2011; Van Laere et al., 2003). Previous reports show that a G to A transition in intron 3 of porcine insulin-like growth factor 2 (IGF2) affects the binding of ZBED6 and significantly up-regulated IGF2 expression in skeletal muscle (Markljung et al., 2009; Van Laere et al., 2003). We predicted the SNP MARC0022311 located in KL intron 1 could change the binding ability of transcription factors including OCT1 by BIOBASE and TFsearch online software (Fig. S1).

The Octamer-binding proteins (OCTs) are a group of highly conserved transcription factors that specifically bind to the octamer motif (ATGCAAAT) and closely related sequences that are found in promoters and enhancers (Zhao, 2013). OCT1 regulates the expression of a variety of genes, including immunoglobulin genes (Dreyfus, Doyen & Rougeon, 1987), β-casein gene (Zhao, Adachi & Oka, 2002), miR-451/AMPK signaling (Ansari et al., 2015), sex-determining region Y gene (Margarit et al., 1998), synbindin —related ERK signaling (Qian et al., 2015).

In the present study, the pGL3-basic was used as the negative control and inserted core promoter fragment with wild-type and mutant-type intron fragments (pGL3-D2-A, pGL3-D2-G). We wanted to check whether there was different in fluorescent activity between two kinds of plasmids (Fig. 2A). The pGL3-D2 was used as control and it did not contain intron fragments, and we wanted to verify whether the transcription factor binding to the inserted intron fragment was the activator or inhibitor (Figs. 2B–2C). The luciferase activity of pGL3-D2-G was significantly higher than pGL3-D2-A (Fig. 2A) and the following OCT-1 RNAi results showed that luciferase activity of pGL3-D2-G significantly decreased in the scrambled and the pGL3-D2-A in PK cells and ST cells (Figs. 2B–2C). The G allele missed one binding sites compared to the A allele (G allele had 2 binding sites, while A allele had 3 binding sites) (Fig. S1), and displayed a higher luciferase activity than A allele (Fig. 2A), suggesting that at this site OCT1 was a repressor. Therefore, we supposed that OCT-1 could bind to the first intron of KL when the SNP was allele A, and then depressed activity of KL promoter.

However, the expression of KL was significantly inhibited after silencing OCT-1. There were several OCT-1 binding sites in porcine KL intron 1 (36,324 bp in length) predicted by BIOBASE and TFsearch online software (Fig. S3A). ChIP analysis showed that OCT-1 interacted with all of three tested regions (1,395 bp to 1,525 bp, 14,322 bp to 14,436 bp, 30,970 bp to 31,141 bp) in PK cells (Fig. S3B). It was possible that there was a synergetic effect between the binding sites. Our aim was to detect the difference of OCT-1 binding sites within two alleles which might change KL gene expression. It was certain that A allele created a novel OCT-1 binding site within the flanking region of MARC0022311 SNP by online prediction (Fig. S1), which was further confirmed by dual-luciferase reporter assay system (Fig. 2) and EMSA (Fig. 4A). In consequence, we hypothesized that OCT1 could dimerise with the chromosome leading to stable binding of the DNA (Tommy et al., 2011; Zabet & Adryan, 2015). In our study, OCT1 could act as an activator and the presence of the third site in the A allele could disrupt the binding of the dimmer leading to lower activity of the A allele.

Klotho physiologically regulates mineral and energy metabolism by influencing the activities of fibroblast growth factors (FGFs) including FGF-2, FGF-19, FGF-23 and their receptors (FGFRs) (Guan et al., 2014; Razzaque, 2009; Wu et al., 2008). Taken together, KL exerts its function via OCT-1 - KL- FGF- FGFR pathway.

Conclusions

In summary, SNP MARC0022311 affected OCT-1 binding ability with the KL promoter. And the KL promoter activity was significantly decreased in allele A of MARC0022311 compared with allele G. Our study indicated SNP MARC0022311 affected porcine KL expression by regulating its promoter activity via OCT-1.

Supplemental Information

Figure S1 Transcription factor binding site prediction of the procine KL intron 1 containing MARC0022311 (KL g.1474 A>G)

Quadrilateral frame indicated the substitutions and extra binding site of OCT-1. (A) Predicted by BIOBASE online software. (B) Predicted by TFserach online software.

Click here for additional data file.

Figure S2 Genotyping results of MARC0022311

(A) PK cells. (B) ST cells. MARC0022311 was marked in gray backgound.

Click here for additional data file.

Figure S3 OCT-1 binding sites in the porcine KL intron 1

(A) Frequency distribution of the predicted OCT-1 binding sites. X-axis indicated the length of the porcine KL intron 1 in bp. Y-axis was the frequency of the predicted OCT-1 binding sites. (B) ChIP analysis of three candidate OCT-1 binding sites (1,395 bp to 1,525 bp, 14,322 bp to 14,436 bp, 30,970 bp to 31,141 bp) in KL intron 1 in PK cells. Primers used for ChIP-PCR was shown in Table 1. Input and R were positive control, while IgG was the negative control.

Click here for additional data file.

Data S1 Raw SNP genotyping results

Click here for additional data file.

We are grateful to Compass Biotechnology Corporation for technical assistance with Illumina SNP analysis. The authors also acknowledge the farmers for providing pig samples.

Additional Information and Declarations

Competing Interests

Author Contributions

Animal Ethics

Data Availability

The authors declare there are no competing interests.

Yan Li and Lei Wang performed the experiments, analyzed the data, contributed reagents/materials/analysis tools, wrote the paper, prepared figures and/or tables, reviewed drafts of the paper.

Jiawei Zhou analyzed the data, contributed reagents/materials/analysis tools, reviewed drafts of the paper.

Fenge Li conceived and designed the experiments, wrote the paper, reviewed drafts of the paper.

The following information was supplied relating to ethical approvals (i.e., approving body and any reference numbers):

All animal procedures were performed according to protocols approved by the Biological Studies Animal Care and Use Committee of Hubei Province, PR China. Sample collection was approved by the ethics committee of Huazhong Agricultural University (No. 30700571 for this study).

The following information was supplied regarding data availability:

The raw data has been supplied as a Data S1.

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
