# Peer review of "Transcription factor organic cation transporter 1 (OCT-1) affects the expression of porcine Klotho (KL) gene"

_PeerJ, doi:10.7717/peerj.2186_

## Round 0.1 · original submission · Major Revisions

The reviewers have found some merit in the manuscript but also feel that a variety of issues need to be addressed prior to the paper being considered for publication in PeerJ.

I therefore strongly urge you to take on board the comments for these reviewers prior to resubmitting the article to PeerJ.

Reviewer 1 ·

Basic reporting

The authors have studied a single promoter of a single gene, whose expression, as they showed, is affected by a particular transcription factor depending on the presence of a specific SNP in its binding site. These results are quite clear, the experimental methodology is solid as far as I can tell, and the paper is in principle at the level and within the scope of this journal. However, the grammar needs to be massively corrected before publication, and the details of the computational analysis need to be clarified, as detailed below:

1) The manuscript needs to be proofread by a professional translator or a colleague with fluent English.

2) The abstract needs to be completely rewritten to address not only grammar mistakes, but also the scientific content. In particular, extensive use of abbreviations in the abstract is not recommended, and if abbreviations are used, each of them needs to be explained.

3) In the abstract, introduction and throughput the manuscript, the authors should use the present tense to describe known facts (in particular, Klotho “is” a membrane protein, not “was”, etc).

4) In the Methods section the authors write: “Transcription factor binding sites were predicted using biological databases (BIOBASE) (http://www.gene-regulation.com/pub/programs.html) with a threshold of 0.90 and TFsearch (http://www.cbrc.jp/TFSEARCH.html) with a threshold of 85.

There are several problems with this. Firstly, BIOBASE is a collective term for several databases and it is not clear which database the authors use, and furthermore, the “threshold” value has no defined meaning in this context. Secondly, I can not comment on the meaning of the threshold in the case of “http://www.cbrc.jp/TFSEARCH.html”, because this page does not exist. The legend and description of Figure S1 should be also extended to explain the use of these databases.

5) Related to the threshold selection in point (4), in the results section the authors surprise the reader with the statement “One hundred and sixty six OCT-1 binding sites were predicted in intron 1 (36324 bp in length) by BIOBASE and TFsearch online software (Fig. 208 S3A).” It is a huge number of “binding sites” for such a small genomic regions, the authors should comment on this. Are these all binding sites, or it just reflects a wrongly selected threshold?

Experimental design

See above

Validity of the findings

See above

Additional comments

See above

·

Basic reporting

I found the reading of this manuscript difficult at times and the authors should improve the stile of the manuscript. In addition, there are several points that need to be clarified:
1. There is no mention in the introduction about MARC0022311 and then the first subsection of the results section is called “MARC0022311 status in pigs”. The authors should include one paragraph in the introduction about this SNP and discuss about previous research done to identify and describe this SNP.
2. “Here we found the SNP MARC0022311 located in KL intron 1 showed a polymorphism in the tested pigs” A SNP is a Single Nucleotide Polymorphism, what did the authors try to say?
3. Figure 2 needs to be better explained. The authors did not clarify how the promoter and intron fragments are positioned in the plasmid. Are they just put one after the other; is there a spacing between them? They should clarify that. They should also clarify to what is the luciferase activity relative to? In the figure caption they should list again the explanation of pGL3-D2-A, pGL3-D2-G and explain that pGL3-Basic is the plasmid without KL-D2 promoter or intron 1 and PGL3-D2 is the plasmid only with KL-D2 promoter. They should explain that the scrambled siRNA was used as control (or even replace scrambled with control siRNA and clarify what sequence they used for that). Finally, why in Fig 2A they show PGL3-Basic and in Fig2B and 2C they show pGL3-D2?
4. Figure 3, the authors need to explain what is the expression relative to (is it WT cells)? They should also clarify the expected band sizes for the WB.
5. “To address whether KL intron 1 contained OCT-1 binding sites in vitro, we used two oligonucleotides (A allele and G allele oligonucleotides) with differing only at SNP MARC0022311 position, as porcine OCT-1 probes in EMSA.” This needs to be rewritten.
6. Figure S2: there is no y-axis and, in addition, each position on the x-axis should be specified.
7. Figure S1 and discussion. The graph is difficult to read. What is actually interesting from that graph is that one middle site for OCT1 (flanked by 2 sites) is removed and this leads to a potential change in expression. Does the G allele display 2/3 of the binding of OCT1 and the expression of the gene in the A allele? Or is there a synergetic effect between the binding sites (see fore example http://www.sciencedirect.com/science/article/pii/S2001037014000142)? This is something the authors should address.
8. These two sentences contradict each other: “In the present study, luciferase activity of pGL3-D2-G was significantly higher than pGL3-D2-A in PK cells and ST cells and the following OCT-1 RNAi results showed that luciferase activity of pGL3-D2-G significantly decreased, confirming OCT-1 was the repressor. Therefore, we supposed that OCT-1 could bind to the first intron of KL when the SNP was allele A, and then depressed activity of KL promoter.” The first sentence needs to be rewritten in a clearer way (also including reference to the Figures they are referring to). The G allele which misses one binding sites compared to the A allele (G allele has 2 binding sites, while A allele has 3 binding sites) displays a higher luciferase activity according to Figure 2A, suggesting that OCT1 is a repressor. If they knock down OCT1, there is no significant de repression of the pGL3 reporter (compare the grey bars and black bars in Figure 2B-2C). At least I am not convinced when looking at the error bars of the middle part of Figure 2B. Then in the second sentence, they claim that binding to allele A (the allele with additional binding sites for OCT1) would lead to “depressed activity”, while it should be the other way around. The more binding sites for a repressor the less activity a gene should display.

Experimental design

Figure 4B and S3B. The ChIP and the input band seem to have the same intensity. The authors should do ChIP with qPCR. In addition, no negative control (a sequence that is not bound by OCT1) is used and, thus, is difficult to judge whether OCT1 really binds to that region. Thus, the authors should also include a negative control.

Validity of the findings

1. Most of the paper is based on the assumption that OCT1 knock down leads to activation of the luciferase assay. In Figure 2B, the authors say that for pGL3-D2-A there is a statistical difference between the OCT1 siRNA and scrambled siRNA. I am looking on the error bars of that barplot and I am not convince that this is true.

2. The authors claim that there are 166 binding sites for OCT1 in the intron1. A single SNP, which changes only 1 binding site, should make a difference for gene regulation? I do not believe that having 165 sites or 166 should lead to any noticeable difference in gene expression. There could be several explanations for this result: the binding sites need to be in accessible DNA in order to be bound by the TF (http://journals.plos.org/plosgenetics/article?id=10.1371/journal.pgen.1001290 or http://nar.oxfordjournals.org/content/43/1/84); the binding of OCT1 could be assisted by co-factors (http://journals.plos.org/plosgenetics/article?id=10.1371/journal.pgen.1003571); OCT1 could dimerise and only the dimer form could lead to stable binding of the DNA. In the last case, OCT1 could act as an activator and the presence of the third site in the A allele could disrupt the binding of the dimmer leading to lower activity of the A allele. There are many explanations, but the one provided by the authors is the least plausible one.

---

## Round 0.2 · Minor Revisions

Thank you for submitting your revised manuscript for our consideration. Thank you for your patience while your manuscript has been reviewed and please accept my apologies for the delay in responding.

It has now been assessed once more by the two original referees, whose comments are copied below. They do have a number of suggestions for improvements/clarifications in the manuscript and would ask you to address these before publication. In particular, please note the comments of Reviewer 2 who observes that some of your responses need to be incorporated into the main text, and who strongly requests that you consider the following additions: a fuller explanation of your ChIP qPCR; more replicates for the luciferase assay; and redoing the computational analysis with a webserver that is available (so readers and reviewers can reproduce the work).

Reviewer 1 ·

Basic reporting

The authors have improved the manuscript, I have only minor corrections which need to be addressed before publication:

1) “MARC0022311 in KL intron 1 appeared a polymorphism (A or G) in Landrace× DIV pigs using the Porcine SNP60 BeadChip and relative luciferase activity of pGL3-D2-G was significantly higher than that of pGL3-D2-A.” --> Needs to be rewritten.

2) “mice has the pattern” --> “mice have the pattern”

3) “Transcription factor binding sites were predicted using biological databases (BIOBASE) (http://www.gene-regulation. com/pub/programs.html) with a threshold of 0.90 and TFsearch with a threshold of 85 (Akiyama, 1995)”. --> The authors did not address my point, which stated that “BIOBASE is a collective term for several databases and it is not clear which database the authors use, and furthermore, the “threshold” value has no defined meaning in this context.” The authors have provided some answer to this point in their response to referees, but this response should be part of the revised manuscript.

4) “Previous reports shows” --> “Previous reports show”

5) “significantly decreased than the scrambled and the pGL3-D2-A in PK cells and ST cells” --> “significantly decreased [in?] the scrambled and the pGL3-D2-A in PK cells and ST cells”

6) “OCT1 could dimerise with the chromosome” --> what does this mean???

7) “was significantly decreased with allele A” --> “was significantly decreased in allele A”

8) “Zabet & Adryan, 2015” is mentioned in the text but not listed in the reference list.

Experimental design

No Comments

Validity of the findings

No Comments

Additional comments

See above

·

Basic reporting

1. I appreciate their explanation of the MARC0022311, but they should include something along those lines in the introduction.
2. They addressed that point.

3.
There was no spacing between them in the Fig2, they were put one after the other. We made pGL3-D2-A and pGL3-D2-G constructs containing intron fragments followed by KL-D2 promoter, then inserted in the downstream of the luc+ gene between the KpnI and HindIII sites (as the pGL3-Basic Vector circle map showed) according to the following reference.
This should be included in the main text.

The luciferase activity was relative to the Renilla luciferase.
This should be included in the Figure caption
In Fig 2A, the pGL3-basic was used as the negative control and inserted core promoter fragment with wild-type and mutant-type intron fragments (pGL3-D2-A, pGL3-D2-G). We wanted to check whether there was different in fluorescent activity between two kinds of plasmids. In Fig 2B and 2C, the pGL3-D2 was used as control and it did not contain intron fragments, and we wanted to verify whether the transcription factor binding to the inserted intron fragment was the activator or inhibitor.
This should be added in the main text. In fact, in the main text the authors do not mention about pGL3-basic. In addition, Figure 2A the comparison is made between the A/G SNP and the negative control and they should be grouped together for easier comparison (one group for PK cells and one group for ST cells).

4.
In Fig 3, the expression of target gene was relative to the expression of β-actin and we used β-actin as the control.
They should add this in the figure caption.

5. They addressed that point.

6.
The X axis represented the position of bases and the Y axis represented fluorescent activity. Besides different color represented different bases. Within reasonable ranges, the higher the fluorescence activity was, the greater the probability of corresponding base was.

They should properly label the axis of the figures. They should also explain this in the Figure caption.

7.
Yes, it was possible that there was a synergetic effect between the binding sites. However, there should be a lot of OCT-1binding sites within KL gene (Fig S3) for both A allele and G allele. Our aim was to detect the difference of OCT-1 binding sites within two alleles which might change KL gene expression. It was certain that A allele created a novel OCT-1 binding site within the flanking region of MARC0022311 SNP by online prediction (Fig S1), which was further confirmed by dual-luciferase reporter assay system (Fig 2) and EMSA (Fig 4A) .
This should be added in the text.

8.
Line 205: We changed this sentence into “In the present study, the luciferase activity of pGL3-D2-G was significantly higher than pGL3-D2-A and the following OCT-1 RNAi results showed that luciferase activity of pGL3-D2-G significantly decreased than the scrambled and the pGL3-D2-A in PK cells and ST cells. The G allele missed one binding sites compared to the A allele (G allele had 2 binding sites, while A allele had 3 binding sites) (Fig S1), and displayed a higher luciferase activity than A allele (Fig 2A), suggesting that at this site OCT1 was a repressor. Therefore, we supposed that OCT-1 could bind to the first intron of KL when the SNP was allele A, and then depressed activity of KL promoter.”
Compared with pGL3-D2, pGL3-D2-A and pGL3-D2-G was inserted the intron fragments. There might be more than one transcription factor binding site. The OCT-1 was the repressor but perhaps other transcription factors were the activator, so overall the luciferase activity increased relative to pGL3-D2 and pGL3-D2-G decreased relative to pGL3-D2-A.
What do they mean by “Compared with pGL3-D2, pGL3-D2-A and pGL3-D2-G was inserted the intron fragments.”
The first sentence should be something like “In the present study, the luciferase activity of pGL3-D2-G was significantly higher than pGL3-D2-A (see Fig 2A) and the following OCT-1 RNAi results showed that luciferase activity of pGL3-D2-G significantly decreased than the scrambled and the pGL3-D2-A in PK cells and ST cells (see Fig 2B-C).”

Experimental design

The authors still did not clarify how much DNA they took for input (they should include this in the text). Also, how many PCR rounds they did? Is the PCR in the linear range? How many biological replicates did they do for the ChIP?

Validity of the findings

Regarding the luciferase assay. I looked at the data and I am still not convinced there is a difference in PK cells. In Figure 2B (pGL3-D2-A OCT1 vs scrambled) the t-test shows a statistically significant difference (p-value=0.036), but the t-test assumes a normal distribution (which the authors did not prove). When I performed a Wilcoxon Rank Sum test (which does not assume a normal distribution anymore), the difference was not statistically significant anymore (p-value =0.57). More replicates might be required to clarify whether the difference is statistically significant or not. Also in Fig 2A the difference between pGL3-D2-A and pGL3-D2-G does not look statistically significant. To prove this, the authors should also use additional negative controls (similar size DNA fragments that would not be bound by OCT-1).

Finally regarding the points raised by reviewer 1 with respect to TFSearch. The authors should redo their analysis with a webserver that works. The manuscript should not be published if one cannot reproduce their computational analysis. The authors should also include in the text the exact tool from Biobase they used and the parameters they selected.

---

## Round 0.3 · accepted · Accept

Thank you for the revised version of the manuscript. The manuscript is suitable for the publication in PeerJ.